# Trend in Research on Characterization, Environmental Impacts and Treatment of Oily Sludge: A Systematic Review

**DOI:** 10.3390/molecules27227795

**Published:** 2022-11-12

**Authors:** Anyi Niu, Xuechao Sun, Chuxia Lin

**Affiliations:** 1International Envirotech Limited, Hong Kong, China; 2Centre for Regional and Rural Futures, Faculty of Science, Engineering and Built Environment, Deakin University, Burwood, VIC 3125, Australia

**Keywords:** petroleum, sludge, industrial waste, pollution, environmental remediation, valorization

## Abstract

Oily sludge is a hazardous material generated from the petroleum industry that has attracted increasing research interest. Although several review articles have dealt with specific subtopics focusing on the treatment of oily sludge based on selected references, no attempt has been made to demonstrate the research trend of oily sludge comprehensively and quantitatively. This study conducted a systematic review to analyze and evaluate all oily sludge-related journal articles retrieved from the Web of Science database. The results show that an increase in oily sludge-related research did not take place until recent years and the distribution of the researchers is geographically out of balance. Most oily sludge-related articles focused on treatment for harmfulness reduction or valorization with limited coverage of formation, characterization, and environmental impact assessment of oily sludge. Pyrolytic treatment has attracted increasing research attention in recent years. So far, the research findings have been largely based on laboratory-scale experiments with insufficient consideration of the cost-effectiveness of the proposed treatment methods. Although many methods have been proposed, few alone could satisfactorily achieve cost-effective treatment goals. To enable sustainable management of oily sludge on a global scale, efforts need to be made to fund more research projects, especially in the major oil-producing countries. Pilot-scale experiments using readily available and affordable materials should be encouraged for practical purposes. This will allow a sensible cost-benefit analysis of a proposed method/procedure for oily sludge treatment. To improve the treatment performance, combined methods are more desirable. To inform the smart selection of methods for the treatment of different oily sludge types, it is suggested to develop universally accepted evaluation systems for characterization and environmental risk of oily sludge.

## 1. Introduction

Globally, petroleum or crude oil has been a major source of energy, as well as a feedstock for producing various petrochemical products for the past decades [1]. The widespread use of petroleum has driven the production of crude oil around the world, with the total annual production of crude oil increasing from 2868 million tons in 1973 to 4141 million tons in 2020 [2]. While the petroleum industry plays a crucial role in economic development, its adverse impacts on the environment are unignorable.

“Oily sludge” is a term collectively used to describe the semi-solid emulsified wastes generated associated with various industrial activities during the exploration, production, transportation, storage, and refining of crude oil. Oily sludge consists of water, petroleum hydrocarbons, and minerals at varying mixing ratios, depending on the source of crude oil, the process through which the oily sludge is generated, and the aging time. According to the origin, oily sludge can be roughly divided into the following types: (a) oilfield pit wastes consisting of drying spilled crude oil being collected during drilling operations and the dumped drilling mud/cuttings, (b) crude oil storage tank sediments, (c) heat exchange bundle cleaning sludge, (d) residues from oil/water separator, (e) sludge of oily wastewater treatment, and (f) activated sludge from biological treatment of oily wastewater. Oily sludge contains potentially toxic petroleum hydrocarbon species, heavy metals, and possibly radionuclides. It has been shown that oily sludge has toxic effects on earthworms [3], affects the growth of plants [4], and could have adverse impacts on human health [5]. Therefore, oily sludge is considered a hazardous material that requires safe handling to minimize its adverse environmental impacts. For this reason, the petroleum industry is legally bound to properly manage oily sludge. Globally, approximately 5 tons of oily sludge is generated from every 1000 tons of crude oil [6]. Therefore, cost-effectively managing oily sludge represents a great challenge to the petroleum industry.

Like other hazardous materials, the strategies for handling oily sludge include containment, detoxification, and elimination [7]. Containment aims to prevent oily sludge from spreading to the surrounding environment by keeping it in storage facilities or immobilizing it via solidification using cementing materials [8]. Detoxification involves the removal of toxic components from oily sludge to reduce its harmfulness via physical separation or chemical/biological degradation [9,10,11]. Elimination is intended to completely destroy the organic components of oily sludge via high-temperature treatment [12]. While these methods are, to varying degrees, effective in minimizing the environmental impacts of oily sludge, they are generally very costly. Consequently, there has been increasing attention paid to the beneficial utilization of oily sludge, mainly including the recovery of fuels and carbonaceous adsorbents from oily sludge, synthesis of composite materials using oily sludge as a feedstock, and miscellaneous uses.

The development of cost-effective technologies for the treatment and valorization of oily sludge requires knowledge of the physicochemical properties of oily sludge and its toxic effects on humans and biota. The characteristics of oily sludge vary due to the difference in its origin. Characterization and toxic assessment of oily sludge are therefore important in providing the knowledge necessary for effectively managing oily sludge generated from the petroleum industry.

For the past decades, there has been increasing research into oily sludge to generate fundamental knowledge and develop innovative technologies for better management of oily sludge. Several research groups have summarized the state of knowledge on oily sludge with a focus on the treatment technologies for either minimizing the environmental impacts of oily sludge or beneficially utilizing oily sludge for fuel or material recovery [6,7]. While these review articles provide the overview of specific subtopics supported by some selected published reports, there has been no effort to evaluate the trend of oily sludge research in a quantitative and comprehensive way. In this study, we analyzed all oily sludge-related journal articles available from the Web of Science database to cover all aspects of oily sludge research, including formation, characterization, environmental impacts, harmfulness reduction treatment, and beneficial utilization of oily sludge. This study aimed to systematically evaluate the research trend of oily sludge by covering all aspects of oily sludge research. In addition, statistical analysis was performed to quantitatively understand the temporal evolution and hot areas of oily sludge research as well as the geographical distribution of the researchers working in this research field.

## 2. Overall Trend

A total of 379 journal articles retrieved from the Web of Science were analyzed and evaluated. The first paper related to oily sludge dates back to 1993, which concerned the dewatering of oily sludge [13]. From 1993 to 2005, the annual number of published journal articles related to oily sludge was no more than six. The number then tended to grow gradually, with a sharp increase since 2019; the number of oily sludge-related journal articles published in 2021 reached 62 (Figure 1a). This reflects that increasing attention has been paid to the environmental issues associated with oily sludge generated from the petroleum industry, which triggers the effort to develop management strategies and treatment technologies for tackling this problem.

The research papers published before 2000 hardly considered the beneficial uses of oily sludge. The research interests during this period focused on the issues related to the disposal of oily sludge, including characterization of oily sludge, dewatering to reduce the volume of oily sludge, solidification to immobilize oily sludge, landfarming to dispose of oily sludge while allowing the petroleum hydrocarbons to be decomposed by the degrading microbes, and incineration of oily sludge, etc.

Of the 379 articles published by July 2022, 96% were research articles, including 45% of the articles with a focus on the treatment for reducing the hazardousness of oily sludge, followed by valorization treatment of oily sludge (40%). Articles dealing with the formation and characterization of oily sludge and the environmental impact assessment of oily sludge only accounted for 9% and 2% (Figure 1b). However, it must be pointed out that information on the characteristics and environmental impacts of oily sludge was covered in some articles focusing on the treatment of oily sludge for either harmfulness reduction or beneficial uses.

Figure 2 shows the inter-citation network of oily sludge-related articles published in the journals with a total number of citations over five times for all relevant articles published in that journal. Journal of Hazardous Materials was the journal receiving the greatest number of citations (1469 times), followed by Bioresource Technology (412 times) and Fuel (375 times). This is consistent with the prevailing research interest in the harmfulness reduction treatment of oily sludge and the beneficial utilization of oily sludge, particularly for fuel recovery. The international collaboration network of oily sludge-related research (Figure 3) indicates that China is leading oily sludge research. There are 249 journal articles on oily sludge that involve co-authors from at least one Chinese research institution. Other key players include Iran (38 articles), India (24 articles), and Canada (20 articles).

## 3. Formation and Characterization of Oily Sludge

Different types of oily sludge are formed under different conditions. So far, only a handful of journal articles have dealt with the formation of oily sludge with a focus on oil storage tanks. Tank bottom sediments are formed due to the precipitation of heavier petroleum hydrocarbons and clays from crude oil. Jiang et al. [14] proposed a method to predict the deposition of sludge in crude oil storage tanks. Farzaneh-Gord et al. [15] found that the exterior surface paint color of the oil storage tank had an impact on the formation of sludge via its effect on the temperature within the oil storage tank. Hassanzadeh et al. [16] examined the effects of different chemical and physical factors, such as surfactants, solvents, temperature, pressure, and mixing conditions, on the formation of oil storage tank bottom sediments. Chen et al. [17] investigated the formation of polymer-containing oily sludge produced during wastewater treatment in offshore oilfields. Unlike oily sludge generated in oil storage tanks or from crude oil processing, oily sludge in oilfield waste pits is formed via a strong interaction of the dumped oily wastes with natural environmental factors such as solar radiation, atmospheric deposition, and underlying soils. Under such environmental conditions, volatile short-chain petroleum hydrocarbons in oily wastes tend to evaporate rapidly. Photocatalytic degradation of petroleum hydrocarbons caused by solar radiation could also occur [18,19].

The characteristics of oily sludge are largely determined by the physicochemical properties of the crude oil from which the oily sludge is derived, as well as the type of oily sludge. Oilfield pit wastes usually contain more solid materials, while sludge from oily wastewater treatment tends to have a high moisture content. Aging of oily sludge results in a decrease in moisture content. Open disposal of oily sludge leads to an increase in solid components as a result of dust addition and a decrease in petroleum hydrocarbon content due to the evaporation of volatile components and partial decomposition via solar radiation-driven chemical oxidation and biodegradation. The major physicochemical parameters of oily sludge that are important for assessing its environmental impacts and providing technical information for selecting suitable treatment methods include pH, electrical conductivity (EC), total solid, density, chemical oxygen demand (COD), total organic carbon (TOC), petroleum hydrocarbon composition, element composition, especially heavy metal(loid)s and nutrient elements such as nitrogen and phosphorus. Jerez et al. [20] developed a method to comprehensively characterize oily sludge. The first step of their analytical procedure involves the separation of the aqueous phase from the oily sludge by centrifugation, followed by the desorption of the oily phase from the solid phase using organic solvents. The aqueous phase sample was used to determine pH, COD, TOC, soluble metals, ammonium, phosphate, etc., while the oily phase sample was used to determine total petroleum hydrocarbon (TPH) and various hydrocarbon fractions; the solid phase sample was analyzed to determine various heavy metal(loid)s and minerals.

The pH of oily sludge was usually between 5.5 to 8 (Table 1) [20,21], although extremely low pH was encountered in sulfur-enriched sludge [22]. Several articles focused on the characterization of the oily phase using different fractionation techniques [23,24,25,26,27]. Kriipsalu et al. [28] carried out a 1-year monitoring of the oily sludge from a flocculation-flotation unit of a wastewater treatment system in a refinery in Sweden. They found that the chemical composition of the sludge was temporally variable. The most abundant petroleum hydrocarbons were nonpolar aliphatic hydrocarbons, while naphthalene, fluorene, and phenanthrene dominated polycyclic aromatic hydrocarbons (PAHs). Xylenes were the major species in the benzene, toluene, ethylbenzene, and xylene (BTEX) group.

The elevated level of heavy metal(loid)s also a major environmental concern. Therefore, appropriate methods for the determination of oily sludge-borne heavy metal(loid)s are important for evaluating the environmental risk of oily sludge. Scharf et al. [29] used an ultrasound-assisted extraction method for the determination of metals in oily sludge using inductively coupled plasma-mass spectrometry (ICP-MS). Wang et al. [30] developed a novel method based on the laser-induced breakdown spectroscopy (LIBS) technique coupled with wavelet transform-random forest (WT-RF) to determine metals in oily sludge samples. Table 1 shows the concentration range of common heavy metals present in oily sludge, including zinc (Zn), copper (Cu), lead (Pb), chromium (Cr), nickel (Ni), cadmium (Cd), and mercury (Hg). It can be seen that the concentration of oily sludge-borne heavy metals varies markedly, with the reported value ranging from 15.4 to 12,249 mg/kg for Zn, 22.6 to 4420 mg/kg for Cu, 40 to 850 mg/kg for Pb, 14 to 200 mg/kg for Cr, 15.2 to 2700 mg/kg for Ni, 7.6 to 100 mg/kg for Cd, and 2.1 to 3766 mg/kg for Hg [31,32,33,34]. Due to its high toxicity, Hg has received more attention, with many journal articles focusing on the quantification and fractionation of oily sludge-borne Hg [34]. Oily sludge could also contain radionuclides. Salih et al. [35] investigated radium (Ra) in 55 oily sludge samples collected from different sites. They found that the levels of Ra were five times the guideline values for soil and regular sludge.

Articles falling within the category of “formation and characterization of oily sludge” also include subtopics on the mobility of environmentally significant elements in the solidified oily sludge and ash produced by incineration or pyrolysis of oily sludge [36,37], as well as the evolution of sludge-borne nitrogen and sulfur during thermal treatment and the associated emission of gaseous pollutants [38,39].

**Table 1 molecules-27-07795-t001:** pH, the concentration of heavy metals, and radioactivity of radionuclides in oily sludge.

Parameter	Range	References
pH	5.5–8.0	[20,21]
Zn (mg/kg)	15.4–12,249	[21,28,31,32,33]
Cu (mg/kg)	22.6–4420	[21,28,31,33]
Pb (mg/kg)	40–850	[21,28,31,32,33]
Cr (mg/kg)	14–200	[21,31,33]
Ni (mg/kg)	15.2–2700	[21,28,31,32,33]
Cd (mg/kg)	7.6–100	[32]
Hg (mg/kg)	2.1–376.6	[28,31,34]
Ra (Bq/kg)	150–1000	[35]

## 4. Environmental Impacts of Oily Sludge

Like crude oil and petroleum fuels, the petroleum hydrocarbons, especially PAHs and heavy metal(loid)s contained in oily sludge, are harmful to ecosystems and human health. However, oily sludge contains more long-chain hydrocarbons, such as asphaltene and resin, which are less degradable and have lower mobility compared to short-chain petroleum hydrocarbons. On the other hand, the enrichment of mineral solids and the addition of chemicals are likely to increase the levels of heavy metal(loid)s that are not degradable. Oily sludge may also contain elevated levels of radionuclides. Research focusing on assessing environmental risk from oily sludge has been limited, with only 2% of the journal articles dealing with this subtopic (Figure 1b).

Wang et al. [40] tested the toxicity of aged petroleum sludge to earthworm Eisenia fetida using mortality, growth, cocoon output, juvenile production, and avoidance behavioral response as test endpoints. They found that the toxic effects were concentration-dependent. Reinecke et al. [3] used the earthworm Eisenia andrei (Oligochaeta) as a test organism to assess the ecotoxicity of oily sludge-containing soils in a landfarming site. They found that low sludge concentrations detrimentally affected earthworm biomass and reproduction.

ElNawawy et al. [4] conducted a plant growth experiment to observe the growth performance and elemental content of three different plant species grown on oily sludge-containing soil at a landfarming site. They found that different plant species responded differently to the land-farmed oily sludge, with ryegrass being more tolerant to the sludge-derived toxicity compared to oats and barley. The uptake of heavy metals by the test plants was not remarkable except for Zn in barley.

Hejazi et al. [5] conducted a field experiment to assess the human health risk from oily sludge in landfarming sites under arid conditions. They found that volatilization of oily sludge-borne hydrocarbons resulted in a significantly high concentration of airborne volatile organic compounds (VOCs) in the atmosphere, which could cause serious human health risk to the onsite workers in the early phase of the landfarming process.

To allow practical applications, operationally defined criteria were proposed for environmental risk assessment of oily sludge, particularly heavy metals in original oily sludge or the sludge residues from either pyrolytic or incineration treatments. Li et al. [41] developed a prediction method for assessing the risk level of heavy metals in oily sludge using laser-induced breakdown spectroscopy (LIBS) combined with hybrid variable selection. They determined the LIBS spectra of 30 oily sludge samples and calculated the corresponding Nemerow index. Mutual information-variable importance measurement (MI-VIM) was proposed for LIBS spectra. A random forest (RF) model was established on the basis of the optimized model parameters and selected feature variables to predict the environmental risk associated with the oily sludge-borne heavy metals. It was concluded that LIBS combined with MI-VIM-RF is an effective method to predict the Nemerow index of oily sludge. Gao et al. [42] evaluated the potential ecological risk of heavy metals in the oily sludge and its pyrolysis char using ecological risk index values. Wang et al. [43] used Risk Assessment Code (RAC) to assess the ecological risk of heavy metals in the sludge incineration ash. Wang et al. [44] used the potential ecological hazard coefficient (E r) to evaluate the pyrolysis residue-borne heavy metals. They found that with the increase of pyrolysis temperature, the comprehensive ecological hazard index (RI) of heavy metals in the residue gradually decreased.

## 5. Harmfulness Reduction Treatment of Oily Sludge

Traditionally, oily sludge was disposed of in landfills, which is costly and could cause air, soil, and water contamination [45]. Landfarming allows the gradual degradation of petroleum hydrocarbons contained in oily sludge and crop production simultaneously and thus reduces the cost associated with oily sludge disposal [46,47]. However, the oily sludge-borne potentially toxic metals are not degradable, and they could accumulate these metals in plant tissues in addition to air, soil, and water contamination. To minimize the environmental impacts of oily sludge, active treatment methods such as stabilization of harmful substances via solidification of oily sludge by cementing agents, incineration of oily sludge to destroy organic pollutants, chemical degradation of oily sludge by advanced oxidation technologies, accelerated biodegradation of oily sludge by bioreactors are desirable [6]. The first step for the harmfulness reduction treatment of oily sludge is to reduce the volume of sludge by dewatering [48].

Within the 167 journal articles dealing with the harmfulness reduction treatment of oily sludge, 61% of them concerns bioremediation, followed by dewatering (11%), chemical degradation (8%), incineration (7%), and solidification (5%). Other papers that do not fall into the above sub-categories are related to oily sludge disposal, transportation of oily sludge, equipment for oily sludge treatment, oil storage tank cleaning, and miscellaneous methods for separation of aqueous, oily, and solid phases with no goal to recover fuels and materials from the treatments (Figure 4).

### 5.1. Dewatering of Oily Sludge

Dewatering or dehydration is the process through which water is removed from the sludge. This can be viewed as a pre-treatment of the original oily sludge to reduce the volume of sludge for disposal or further harmfulness reduction and valorization treatments. Sludge dewatering can be achieved by different methods, including (a) natural drying, (b) thermal drying, (c) centrifugation, (d) electro-kinetic dewatering, (e) demulsification, (f) chemical conditioning, and (g) freezing/thawing.

Natural drying of oily sludge is very slow. Dewatering of oily sludge can be accelerated by heating oily sludge. Several research groups investigated the effect of fry-drying on the dehydration of oily sludge [49,50]. Gao et al. [51] applied hydrothermal treatment combined with in-situ mechanical compression to enhance the dehydration of oily sludge. Demulsification using surfactants enhances the separation of the aqueous and oily phases, allowing rapid dewatering of oily sludge. Long et al. [52] reported the dewatering of floated oily sludge by a bio-surfactant rhamnolipid. Xu et al. [53] found that oily sludge dehydration can be markedly enhanced by microwave irradiation. Yang et al. [54] conducted bench-scale experiments to investigate the removal of water from oily sludge using electro-kinetic (EK) techniques. However, the application of the EK method for oily sludge dewatering may not be cost-effective unless oil recovery can also be simultaneously achieved. The treatment method that attracts more research interest appears to be chemical conditioning, with over 31% of articles on oily sludge dewatering dealing with this subtopic. The commonly used conditioners include lime, aluminum salts, ferric iron salts, and polyelectrolytes [55]. Guo et al. [56] found that acidic conditions favored the dewatering of oily sludge due to enhanced aluminum solubility. Hwa and Jeyaseelan [57] used industrial waste fly ash as a conditioner to reduce the cost of treatment. Dehydration of oily sludge can be achieved by centrifugation [58]. However, centrifugation alone may be ineffective. Mao et al. [59] investigated the effects of ultrasound-assisted centrifugation on the dewatering of oily sludge. The freezing/thawing method is more appropriate for simultaneous dewatering and oil recovery [60].

### 5.2. Biological Treatment

The biological treatment of oily sludge involves using microorganisms to decompose petroleum hydrocarbons in oily sludge. Oily sludge consists predominantly of long-chain hydrocarbons. Consequently, the decomposition of oily sludge-borne petroleum hydrocarbons by naturally occurring microbes is usually slow. To accelerate the degradation process, bioaugmentation, i.e., the addition of petroleum hydrocarbon-degrading microbes into oily sludge, is necessary [61]. A large number of journal articles focus on the isolation, identification, and assessment of bacteria that have a strong capacity to decompose oily sludge-borne hydrocarbons [62,63,64]. However, petroleum hydrocarbon-degrading fungal species have been receiving increasing attention [65]. The rate of petroleum hydrocarbon decomposition also depends on environmental conditions such as temperature, moisture content, pH, nutrient status, presence of toxic substances, etc., that affect the growth of petroleum hydrocarbon degraders. In order to enhance the microbially mediated degradation of petroleum hydrocarbons, bio-stimulation, i.e., the creation of favorable environmental conditions, especially the addition of nutrients to promote microbial growth, is required [61].

While landfarming is low-cost and allows the treatment of large amounts of oily sludge, this treatment method is relatively inefficient. It could have marked off-site adverse impacts on the surrounding environment [6]. In comparison, active treatment systems with optimized reaction conditions, such as biopiles or bioreactors are more efficient and environmentally friendly. Biopiles are usually constructed with a liner to prevent oily sludge-borne contaminants from infiltrating into the underlying soil and shallow aquifers, minimizing soil and groundwater contamination. The more favorable conditions, such as higher temperature, appropriate moisture content, and sufficient supply of nutrients in these active treatment systems, enable rapid growth of petroleum hydrocarbon degraders [66]. The biodegradation of oily sludge-borne petroleum hydrocarbons can be further enhanced by composting that introduces bulking agents such as soil and biomass into the reaction systems to improve carbon to nitrogen ratio (C/N) and aeration status [6,18]. Composting degradation of oily sludge can be operated in either biopiles or bioreactors [66,67]. Apart from composting treatment, soil slurry bioreactor systems have also received increasing research attention [68].

The use of surfactants can enhance the interaction between the microbes and the petroleum hydrocarbons for oily sludge treatments [69]. There have been increasing research interests in bio-surfactants produced by soil microbes and their effects on enhancing the biodegradation of oily sludge-borne hydrocarbons [70,71]. Other strategies to accelerate the bio-treatment process include the pre-treatment of oily sludge by chemical oxidation to convert the long-chain hydrocarbons into small organic molecules to facilitate the subsequent biodegradation [72].

### 5.3. Solidification

This treatment method involves the use of curing agents to solidify oily sludge in order to stabilize the sludge-borne harmful substances. The curing agents used for oily sludge solidification include Portland cement [73], carbonaceous fly ash [74], and phosphogypsum [75]. Zhang et al. [8] used vinyl benzene, styrene, butyl acrylate, and persulfate as raw materials to prepare a new type of sludge-solidified polymer through suspension and solution polymerization. Qin et al. [76] solidified oily sludge using molten blast furnace slag, which allowed the oily sludge-borne hydrocarbons being rapidly decomposed while the sludge-borne heavy metals were immobilized.

While solidification of oily sludge reduces the mobility of sludge-borne toxic substances, there is no guarantee that the stabilized toxic materials are not released from the solidified sludge, especially after the solidified sludge is weathered or in contact with corrosive agents. For this reason, the majority of journal articles focus on evaluating the leachability of heavy metals from solidified sludge.

### 5.4. Chemical Oxidation

Chemical oxidation treatment of oily sludge involves the use of oxidants to chemically decompose oily sludge-borne hydrocarbons. Advanced oxidation process (AOP) based on the Fenton reaction theory is a frequently used method for chemically treating oily sludge [77]. The effectiveness of the Fenton-driven decomposition of oily sludge-borne hydrocarbons can be enhanced by combining other technologies, such as ultrasonic extraction [78] and surfactant enhancement [79]. Ozone can also be used as a potent oxidant to decompose oily sludge-borne hydrocarbons [80]. Sun et al. [81] developed a microbubble ozonation method to improve the effectiveness of the oily sludge treatment.

There has been increasing research interest in the use of hydrothermal methods for treating oily sludge [82,83]. Some of these methods could potentially be used for decomposing oily sludge-borne hydrocarbons. Wet air oxidation treatment using oxygen as an oxidant under high temperature and high pressure (subcritical conditions) are capable of decomposing oily sludge-borne hydrocarbons by free radicals generated in the systems [84]. Zhao et al. [85] developed a two-stage wet air oxidation procedure for treating oily sludge. Shi et al. [86] treated the oily sludge using a thermal hydrolysis oxidation process. Supercritical water can also be used as the reaction medium under higher temperature and pressure conditions compared to wet air oxidation to achieve a higher level of hydrocarbon oxidation [87,88,89]. However, the hydrothermal method is usually very expensive because special metal alloys are required to make the reactors strong enough to withstand high temperature and pressure conditions. To overcome this drawback, catalysts such as Ni^2+^ can be used to lower the reaction temperature and pressure while achieving even better hydrocarbon decomposition outcomes [90,91].

### 5.5. Incineration

This treatment method is intended to completely decompose the oily sludge-borne organic components by allowing the combustion of the oily sludge with a sufficient supply of oxygen [4]. The factors affecting the performance include but are not limited to the physicochemical properties of the oily sludge, type of incinerators, combustion temperature, type of auxiliary fuels, and length of treatment time. Many researchers have attempted to optimize the operating conditions to improve the efficiency of treatment [92,93,94].

Co-combustion of oily sludge with biomass improves the treatment efficiency [95]. Therefore, there have been increasing research interests in this aspect. Liu et al. [96] investigated the characteristics and kinetics of the co-combustion of the oily sludge with litchi peels using thermogravimetric analysis. Cao et al. [97] examined the effect of cornstalk addition on the combustion characteristics of oily sludge. Apart from biomass, some researchers also investigated the co-combustion of oily sludge and coal [98].

### 5.6. Other Harmfulness Reduction Treatment-Related Articles

There are a few articles that do not fall into any of the above sub-categories. These include papers with a focus on the methods for oily sludge disposal [99], transport of oily sludge [100], oil tank cleaning [101], electro-demulsification of oily sludge [102], life-cycle assessment of oily sludge treatment approaches [103], etc.

### 5.7. Comparison among the Different Treatment Approaches

As described above, there are five major approaches that can be used to minimize the environmental impacts of oily sludge: (a) dewatering, (b) biodegradation, (c) solidification, (d) chemical degradation, and (e) incineration. Although the cost-effectiveness of individual methods within each approach varies, common limitations of each treatment approach can be identified (Table 2).

Strictly speaking, dewatering does not lead to the hazardousness reduction of oily sludge but just reduces the total volume of oily sludge that needs to be handled. Therefore, this treatment approach only serves as a pre-treatment process for the follow-up disposal in waste storage facilities or further treatments to immobilize or eliminate the hazardous materials contained in the oily sludge. Although the costs for biological treatment are generally low, it takes a long period of time to decompose the majority of the petroleum hydrocarbons contained in oily sludge, and some of the long-chain hydrocarbons cannot be practically degraded by microbes within a reasonable time frame. Furthermore, biodegradation is not favored under dry or cold climatic conditions where long-term maintenance of manipulated environmental conditions (temperature or/and freshwater supply) for microbial degraders could make the treatment practice extremely expensive. Solidification, chemical degradation, and incineration are all highly effective and can be performed within a short period of time. However, the costs associated with these treatments are generally high. For this reason, solidification may only be used for treating small amounts of oily sludge when there is a need to urgently meet legal requirements. Chemical treatment may be more appropriate to be used as a pre-treatment of oily sludge for follow-up biodegradation.

## 6. Valorization Treatment of Oily Sludge

Valorization treatment is intended to recover fuels or/and valuable materials from oily sludge. It has been reported that oily sludge was directly used as a feedstock for making construction materials such as tiles, roadbed materials, and soil aggregates [104,105,106]. However, such beneficial uses without appropriate treatment may pose a certain level of environmental risk. Valorization treatment of oily sludge for fuel recovery requires the separation of the oily phase from the aqueous and solid phases. This can be performed by different methods, mainly including centrifugation extraction, ultrasonic extraction, solvent extraction, surfactant treatment, hydrothermal treatment, anaerobic co-digestion, bioelectrochemical treatment, freezing/thawing treatment, froth flotation, pyrolysis. Among these methods, pyrolytic treatment of oily sludge has received the most research interest, with over 50% of the articles concerning oily sludge valorization focusing on the pyrolysis of oily sludge. This is followed by surfactant treatment (14%), hydrothermal treatment (10%), solvent extraction (9%), and ultrasonic extraction (5%). Other treatment methods and uses of original oily sludge and treatment residues only account for a small proportion of the articles (Figure 5).

### 6.1. Ultrasonic Extraction

Apart from being used for the dewatering of oily sludge, ultrasonic extraction can also be exploited to separate the aqueous, oily, and solid phases of oily sludge, aiming at the recovery of petroleum hydrocarbons from the sludge [3]. Several authors have investigated the factors that affect the rate of oil recovery to optimize the operational conditions for oil recovery [107,108,109].

While technically, oil recovery can be achieved by ultrasonic extraction, but using this method alone for oil recovery is not cost-effective. It is, therefore, more appropriate to use ultrasonic extraction in conjunction with other methods to achieve a better oil recovery effect. Zhang et al. [110] combined ultrasonic extraction with the freezing/thawing method for oil recovery from oily sludge. Gao et al. [111] investigated the joint effects of surfactant separation and ultrasonic extraction on the removal of oil from the sludge. Hu et al. [112] evaluate the effect of ultrasound-assisted solvent extraction on oil recovery from oily sludge.

### 6.2. Solvent Extraction

This method uses extracting solvents to solubilize oily sludge-borne hydrocarbons, which allows the separation of the oily phase from the aqueous phase and solid phase through stratification. The oil dissolved in the solvents can then be recovered via distillation [113]. The recovery rate of oil from oily sludge by solvent extraction is affected by many factors, such as the solvent type, the solvent concentration, the ratio of solvent to sludge, temperature, speed of rotation, treatment time, and pH. Different solvents have been tested for their effectiveness in oil recovery. These include methyl-ethyl ketone, liquefied petroleum gas condensate, tertiary amines, sodium lignosulfonate, cyclohexane, n-hexanol, n-butanol and kerosene, xylene, toluene, n-octane, etc. [114,115,116,117,118,119].

To optimize the operational conditions for better oil recovery, Zubaidy and Abouelnasr [114] compared the effect of solvent/sludge ratios on the oil recovery rate. They found that a ratio of 4:1 allowed optimal oil recovery from the oily sludge by methyl ethyl ketone and liquefied petroleum gas condensate. Dai et al. [116] found that protonated tertiary amines promoted the detachment of crude oil from the sludge surface, and a hybrid process with tertiary amines and protonated tertiary amines improved the recovery of oil from the treated sludge. Ma et al. [118] examined the effects of temperature, solvent concentration, rotation speed, treatment time, and pH on the oil recovery rate. They found that the optimal temperature, solvent concentration, rotation speed, treatment time, and pH was 30 °C, 2.0 g·L^−1^, 200 rpm, 60 min; and 11, respectively. Al Zubaidi [120] proposed a multi-stage extraction procedure to maximize oil recovery. The separate experimental results by Tian [121] and Hou et al. [119] all suggested that ionic liquids could effectively decrease the heavy oil interaction force. Therefore, the presence of ionic liquids can effectively improve oil recovery by solvent extraction.

### 6.3. Surfactant Treatment

The function of surfactants as demulsifiers for dewatering oily sludge can be further exploited to recover crude oil. Unlike dewatering, surfactant treatment aiming at the recovery of oil from the sludge requires simultaneous separation of the oily phase from the aqueous phase and solid phase. Liang et al. [122] investigated the solid effects on surfactant-driven de-oiling of oily sludge. There is a wide variety of surfactants that can be used for oil recovery from oily sludge [113]. To achieve cost-effective oil recovery using surfactant treatment, most of the research interests have been directed to evaluate the cost-effectiveness of surfactants and optimize the operating conditions for achieving a high oil recovery rate [122,123,124,125,126]. Some researchers further explored the possibility of improving the oil recovery rate from special oily sludge types by combining surfactant treatment with other methods. For example, Zheng et al. [127] investigated simultaneous dewatering and oil recovery from high-viscosity oily sludge through a process combining demulsification, viscosity reduction, and centrifugation. Yao et al. [128] used a combined surfactant and adsorption method for the recovery of petroleum hydrocarbon from oily sludge.

Owing to the possible secondary pollution caused by chemical surfactants, there have been increasing research interests in developing and applying biosurfactants that are less toxic and biodegradable to replace chemical surfactants for oily sludge valorization treatment [113,129]. Some researchers have made efforts to isolate, identify and characterize biosurfactant-producing bacteria [130]. So far, rhamnolipids are the biosurfactant that has attracted more research interest because they can be produced by petroleum hydrocarbon-degrading microbes (*Pseudomonas aeruginosa*) [131,132]. Pornsunthorntawee et al. [133] also showed a slight improvement in oil recovery from 51–55% (chemical surfactants) to 57–61% (biosurfactants).

### 6.4. Hydrothermal Treatment

Apart from being used to dehydrate and decompose oil sludge-borne petroleum hydrocarbons, hydrothermal treatment methods can also be used for recovering fuels from the sludge. The methods that can be potentially used for fuel recovery include pressurized hot water extraction (PHWE), thermal hydrolysis (TH), wet air oxidation (WAO), hydrothermal carbonization (HTC), hydrothermal liquefaction (HTL), supercritical water upgrading (SCWU), supercritical water oxidation (SCWO), and supercritical water gasification (SCWG) [85]. The efficiency of supercritical fluid extraction is affected by the physicochemical characteristics of the to-be-treated oily sludge as well as the type of supercritical fluid, ratio of supercritical fluid to sludge, and operating conditions, especially temperature and pressure that are crucial in maintaining a supercritical state [113]. Some researchers explored the possibility of using supercritical water for the recovery of liquid fuels [134,135,136,137,138]. More recently, the focus has been placed on the production of syngas from oily sludge using subcritical and supercritical water [139,140,141,142]. To facilitate fuel recovery, some researchers also examined the effects of catalysts on the oil recovery and gasification processes [143,144].

In addition to supercritical water, other supercritical fluids were also used for fuel recovery from oily sludge. Ávila-Chávez [145] used supercritical ethane to extract the sludge-borne hydrocarbons and found that the oil recovery rate could reach nearly 60% under optimal conditions. The experimental results obtained by Ma et al. [146] showed that an even higher oil recovery rate was achieved via supercritical CO_2_ extraction.

### 6.5. Pyrolytic Treatment

Pyrolytic treatment methods involve the thermal treatment of oily sludge in the presence of limited oxygen. This process allows the recovery of both liquid and gaseous phase fuels from the oily sludge while simultaneously producing chars that can be used as adsorbents. The product distribution is affected by the physicochemical properties of oily sludge and the operating parameters set for the pyrolysis of oily sludge, which mainly includes pyrolysis temperature, heating rate, duration of pyrolysis, and reaction atmosphere [147]. Consequently, many researchers have made efforts to examine the effects of factors on the pyrolysis of various oily sludge and the characteristics of the pyrolytic products [148,149,150,151,152].

With increasing pyrolysis temperature, the dominant process evolves from the vaporization of light hydrocarbons at 100–200 °C to desorption and the thermal cracking of light hydrocarbons at 200–380 °C to cracking of heavy hydrocarbons into smaller molecular weight compounds at 380–550 °C [153] with char formation taking place at the final stage of pyrolysis via aromatization, carbonylation, cyclization and polymerization reactions [154]. At the same pyrolysis temperature, an increase in heating rate and pyrolysis duration enhances pyrolytic effects. Large amounts of liquid and gaseous fuels were generated in the CO_2_ atmosphere, compared to the N_2_ atmosphere [155]. So far, research work related to the pyrolysis of oily sludge was frequently performed using electrically heated reactors [155,156,157]. There has been increasing research interest in the use of microwave-heated reactors for the pyrolytic treatment of oily sludge [158,159,160], given that microwave-driven heating methods could facilitate the pyrolytic process by accelerating the heating rate of the oily sludge and consequently reducing the processing time [161].

The quality of recovered oil from oily sludge by ordinary pyrolysis is usually low [156]. Catalysts such as alkaline-earth metals and transition metals can be used to overcome this problem. The oxides, chlorides, sulfates, and carbonates of these metals, zeolites, bentonite, and some industrial wastes, can also be used as effective catalysts for the pyrolysis of oily sludge [130,162,163]. Shie et al. [164] showed that the effect of calcium-containing compounds on improving the yield and quality of the recovered oil was in the following decreasing order: CaO > CaCO_3_ > CaCl_2_ > Ca(OH)_2_. Milato et al. [165] used zeolites to facilitate the cracking of the oily sludge-borne hydrocarbons.

Another avenue to improve the recovery of pyrolytic products is the co-pyrolysis of oily sludge with other organic matter. Many researchers have used biomass, particularly those from agricultural biomass [166,167,168,169,170]. Some researchers also used organic substances/wastes from industrial sources such as coal, plastics, and waste tires as the feedstock for the co-pyrolysis of oily sludge [171,172,173].

Of the 76 pyrolysis-related journal articles, most aimed at recovering either liquid or gaseous fuels. In contrast, about 30% of the articles had a focus on the characterization of the char (the solid phase of the oily sludge pyrolytic products) and its potential for being used as adsorbents for environmental remediation, particularly for removal of water-borne pollutants such as heavy metal(loid)s. Other beneficial uses of the pyrolytic residues include being used as part of the feedstock for making supercapacitors or catalysts.

### 6.6. A Comparison of Features of the Valorization Treatment Approaches

While the five valorization treatment approaches mentioned above, all allow recovery of oil from the oily sludge and, therefore, effectively reduce the amount of sludge-borne petroleum hydrocarbons that otherwise need to be immobilized or eliminated to minimize their environmental impacts, each of these approaches has its specific features (Table 3). For ultrasonic treatment, solvent extraction, and surfactant treatments, wastewater and solid wastes are formed after oil recovery. These wastes require additional treatment, especially for the wastes from solvent extraction and surfactant treatment that contain secondary pollutants from the solvents and surfactants. Apart from oil recovery, both hydrothermal treatment and pyrolytic treatment also allow the recovery of fuel gases. Furthermore, chars that can be used as adsorbents for environmental applications rather than solid wastes are formed. For pyrolytic treatment, no wastewater is generated. However, toxic gases may be formed from the pyrolysis of oily sludge.

Due to the high costs associated with the installation of the required equipment and relatively low treatment capacity, ultrasonic treatment is only appropriate for small-scale treatment of oily sludge. In contrast, solvent extraction surfactant treatment and pyrolytic treatment can be applied to the large-scale treatment of oily sludge. The expensive equipment and high operating costs for hydrothermal treatment may limit its application where the required capital investment is unavailable.

## 7. Conclusions and Outlook

Critical analysis and evaluation of the 379 journal articles retrieved from the Web of Science reveal that an increase in oily sludge-related research did not occur until recent years. And the distribution of the relevant researchers is of geographical imbalance. It is surprising that many major oil-producing countries, such as the USA, Russia, Saudi Arabia, Iraq, and UAE, have made a disproportionate contribution to knowledge generation on this topic. It is therefore suggested that more funding from petroleum industry sectors in these countries should be allocated to support research projects dealing with oily sludge.

To date, most of the research findings have been obtained from laboratory-scale experiments, and the cost-effectiveness of the proposed treatment methods is not adequately considered. Many experiments used expensive materials and adopted somewhat unrealistic operating conditions. For practical purposes, pilot-scale experiments using readily available and affordable materials should be encouraged. This will allow a sensible cost-benefit analysis of a proposed method/procedure for oily sludge treatment being performed to evaluate its cost-effectiveness.

Although many methods have been proposed, few of them could achieve satisfactory treatment outcomes when used alone. To improve the treatment performance, combined methods are more desirable. There is, therefore, a need to adopt an integrative approach when it comes to the development of treatment technologies for oily sludge.

While the beneficial use of oily sludge seems to be a more attractive approach, the value of the recovered fuels or/and materials does not necessarily offset the additional costs for the more complex treatment processes. It is, therefore, worthwhile to conduct comparative experiments to evaluate the cost-effectiveness between the valorization treatment and harmfulness reduction treatment in terms of sustainable management of oily sludge.

Selection of the most appropriate method is crucial for achieving cost-effective treatment goals. Given that the cost-effectiveness of a treatment method heavily depends on the physicochemical properties and potential toxicity of the to-be-treated oily sludge, characterization and toxicity assessment of the oily sludge are very important. However, these are the areas that have so far received very limited research attention. Therefore, efforts need to be made to develop universally accepted evaluation systems for the characterization and environmental risk of oily sludge, which can be used to inform the selection of suitable methods for the treatment of oily sludge.

## 8. Materials and Methods

Journal articles on oily sludge were retrieved from the Web of Science as of 30 July 2022. The words “oily sludge” and “crude oil tank sludge” were used for the retrieval of relevant journal articles from this database. Workbooks were established on Vosviewer from data exported from the Web of Science in bibliographic database file format. “Citation” was selected under the “type of analysis”. Under the “unit of analysis”, “country,” and “source” were selected.

The titles of the retrieved articles were initially checked one by one to avoid duplicate inclusion. Five categories were set for the classification of subtopics under the topic of “oily sludge”, including (a) a literature review article, (b) an article focusing on the formation and characterization of oily sludge, (c) an article focusing on environmental impacts of oily sludge, (d) article focusing on harmfulness reduction treatment, and (e) article focusing on valorization treatment. For the Categories “harmfulness reduction treatment” and “valorization treatment”, the further division into sub-categories was made, with Sub-category “pyrolytic treatment” being further broken down into “fuel recovery”, “adsorbents” and “others”. Table 4 shows the details of the classification system.

The initial assignment of each retrieved article to a level 1 category was performed by screening the article title and making a decision based on reliable indicative words. This was followed by checking the article abstract and, if needed, the research objectives and methodology section in the full paper for those articles with insufficient information provided in the article title. Further assignment of an article within either the Category “harmfulness reduction treatment” or Category “valorization treatment” to a sub-category was performed by carefully checking the abstract and if needed, the main text of the article. Where more than two treatment methods were covered in the same article, the full article was reviewed to determine which sub-category it should belong based on the estimated weighting of each treatment method.

VOSviewer 1.6.18 and Microsoft Excel were used to perform statistical analysis of the data collected in this study.

## Figures and Tables

**Figure 1 molecules-27-07795-f001:**
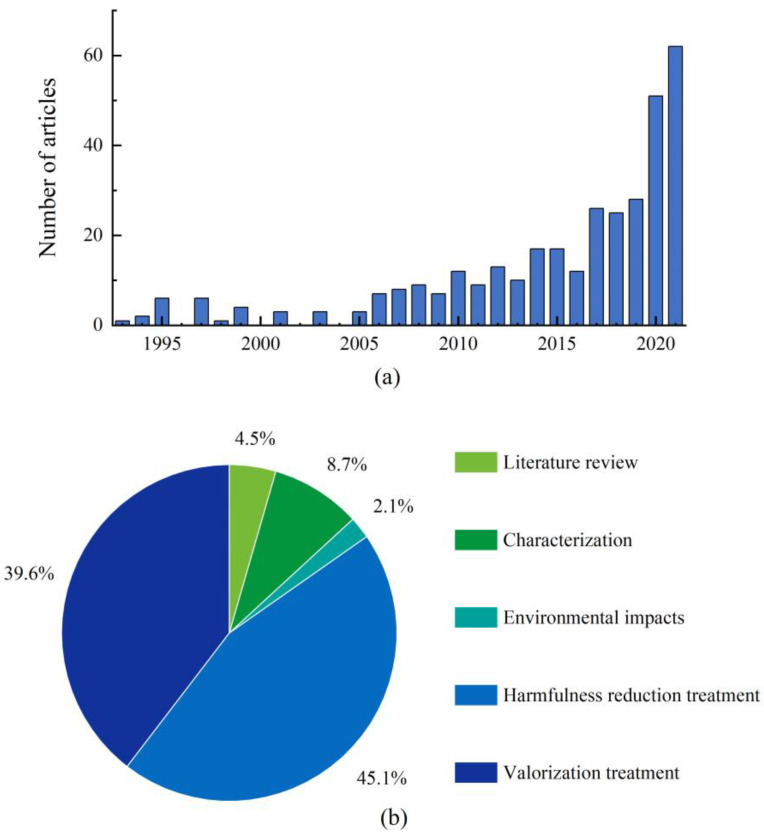
A graph showing the change in the annual number of oily sludge-related journal articles published from 1993 to 2021 (**a**) and a pie chart showing the percent share of 5 journal article categories based on 379 journal articles published by July 2022 (**b**).

**Figure 2 molecules-27-07795-f002:**
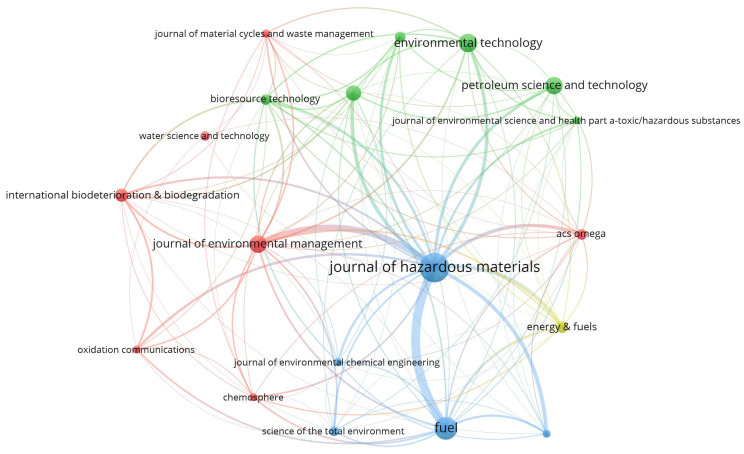
A diagram showing the inter-citation network of oily sludge-related articles published in the journals with a total number of citations more than five times for all relevant articles published in that journal to July 2022. The larger the size of the circle, the higher the number of citations; circles with the same color indicate more frequent inter-citation among the journals.

**Figure 3 molecules-27-07795-f003:**
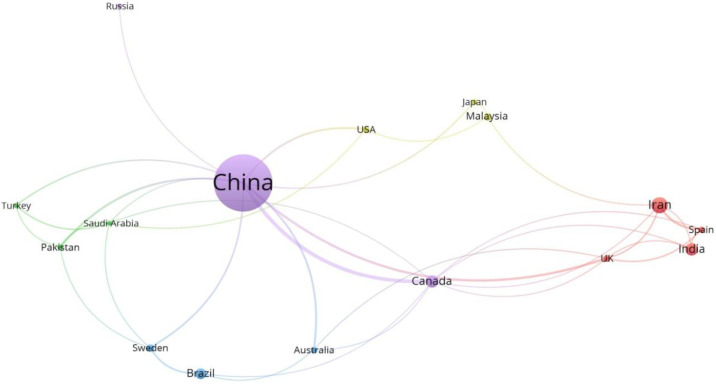
A diagram showing the international collaboration network of oily sludge research. Only countries with more than five articles are included in the statistical analysis. The larger the size of the circle, the higher the number of citations; circles with the same color indicate more frequent collaboration among the countries.

**Figure 4 molecules-27-07795-f004:**
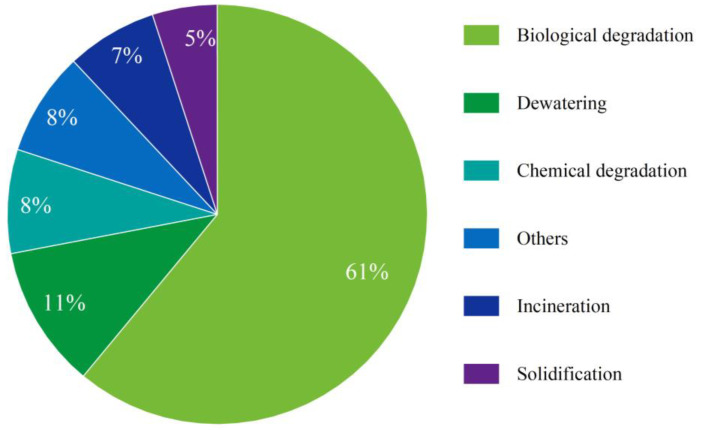
A pie chart showing the percent share of journal articles dealing with different harmfulness reduction treatment methods.

**Figure 5 molecules-27-07795-f005:**
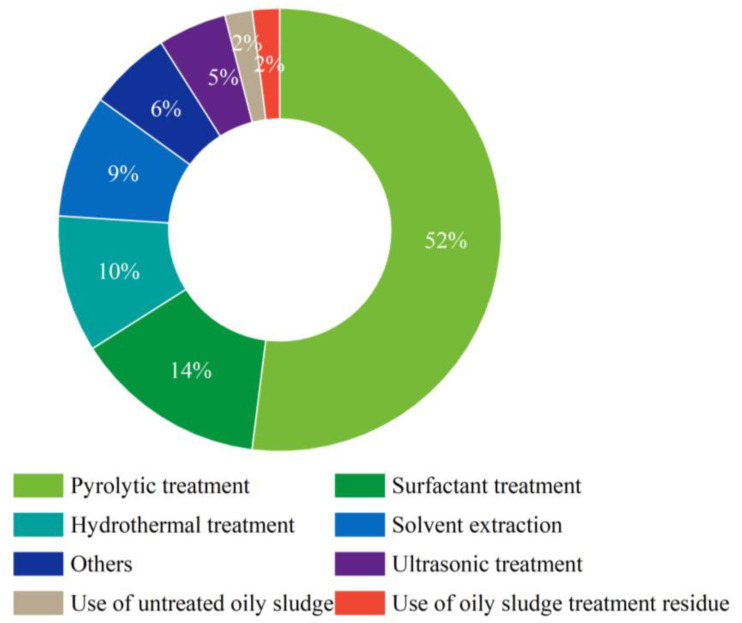
A donut chart showing the percent share of journal articles dealing with different valorization treatment methods.

**Table 2 molecules-27-07795-t002:** A comparison of the features, limitations, and optimized applications among the five major treatment approaches for hazardousness reduction of oily sludge.

Approach	Effectiveness	Duration	Cost	Optimized Application
Dewatering	It only reduces the volume of oily sludge. No toxic materials are removed or eliminated from the oily sludge	Depending on the method selected	Generally Low	Pre-treatment of oily sludge for disposal or further treatment using other approaches
Biodegradation	Some long-chain hydrocarbons cannot be decomposed. The toxicity of oily sludge-borne heavy metal(loid)s cannot be reduced	Time-consuming	Generally low	Only suitable for the treatment of oily sludge containing less long-chain hydrocarbons and under warm and humid climate conditions
Solidification	The mobility of petroleum hydrocarbon and heavy metal(loid)s are effectively reduced, minimizing the environmental risk of the oily sludge-borne hazardous materials	Treatment can be performed within a reasonably short period of time	High	It may only be practical for treating a small amount of oily sludge in order to meet legal requirements for rapidly eliminating the environmental risk
Chemical degradation	Effective decomposition of petroleum hydrocarbons but no effect on reducing the toxicity of heavy metal(loid)s	Treatment can be performed within a reasonably short period of time	High	More appropriate to be used as a pre-treatment for the follow-up biological treatment
Incineration	Complete decomposition of petroleum hydrocarbons. However, toxic gases can be generated. No effect on reducing the toxicity of heavy metal(loid)s	Treatment can be performed within a reasonably short period of time	Extremely high	Only where capital investment is available to install facilities for heat recovery

**Table 3 molecules-27-07795-t003:** A comparison of features among the five major valorization treatment approaches for oily sludge.

Approach	Recovered Products	Waste Products	Disadvantages	Optimized Application
Ultrasonic treatment	Oil	Wastewater Residue	High equipment costs and low treatment capacity	Small-scale treatment of oily sludge
Solvent extraction	Oil	Wastewater Residue	Secondary pollution	Large-scale treatment of oily sludge
Hydrothermal treatment	Oil, fuel gases, hydrochar	Wastewater	High energy and water consumption	Capital is available for investment in equipment
Surfactant treatment	Oil	Wastewater Residue	Secondary pollution	Large-scale treatment of oily sludge
Pyrolytic treatment	Oil, fuel gases, biochar	Waste gases	High energy consumption	Treatment of oily sludge with low water content

**Table 4 molecules-27-07795-t004:** The classification system of the oily sludge-related journal articles was retrieved from the Web of Science database.

Category 1	Category 2	Category 3
Literature review		
Formation and characterization		
Environmental impacts		
Harmfulness reduction treatment	Dewatering	
	Biological degradation	
	Chemical degradation	
	Solidification	
	Incineration	
	Other treatments	
Valorization treatment	Pyrolytic treatment	Fuel recovery
		Adsorbents
		Others
	Solvent extraction	
	Surfactant treatment	
	Hydrothermal treatment	
	Ultrasonic treatment	
	Use of untreated oily sludge	
	Use of oily sludge treatment residues	
	Other treatments	

## Data Availability

Not applicable.

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
