# Peer review of "Trend in Research on Characterization, Environmental Impacts and Treatment of Oily Sludge: A Systematic Review"

_molecules, 2022, doi:10.3390/molecules27227795_

Round 1
Reviewer 1 Report
Manuscript ID molecules-1959204
“Trend in oily sludge research: A systematic review”
The subject of the present manuscript is interesting and presents scientific relevance. However, I missed innovation in the present study, what is innovative about your work compared to other works conducted in this line of research. It is important to emphasize the innovative character of your work. In addition, several aspects must be taken into account during the writing of the text.
Below I indicate some points of improvement / suggestions.
- The abstract should be improved. Address in the abstract what your review will present, as well as what's new in your work.
Introduction: authors should emphasize what are the main differences between their work and previous examples published in the literature, for example https://doi.org/10.1007/s11356-020-11176-2
- How about the search methods used in systematic review? The methodology should be improved.
- Figure 3: The figure 3 is very interesting, but I think it is better put the countries in upper case.
- Figure 4: As suggestion, organizing the figure legend in proportion of % or alphabetical order.
- Figure 5 should be improved
- Figure 5. The authors should mention each article comprised each different valorization treatment methods.
- In my opinion, Figure 6 is not necessary in this manuscript
- References must be formatted according to the journal's rules
- All abbreviations must be cited in the text
- All figures and tables must be cited in the text
- It is missed tables that summarize the studies already carried out in each segment of the present review.
- It is missed the critical point of view in conclusion section.
Author Response
Please see the file attached

Reviewer 2 Report
Overall, it is an interesting review article with abundant and relative new references, but needs a major revision. It is worth to publishing only if the authors can make substaintial improvement according to the following suggestion:
1. The foremost suggestion is to add tables from Chapter 3 to Chapter 6. I am a bit surprised to see a review article with basically just words and so few tables. For example, in Chapter 5, the authors could make a table to compare the different techniques for harmfulness reduction of oily sludge (condition of use, efficiency, target objective/material, pros and cons, etc.), and similar suggestion to the other Chapters. The authors should keep in mind on how to contribute to the academia and how the readers could benefit from your article in the best way; tables and figures can help and make your article not only more readable but also more valuable. Also, figures with just statistical values may not be able to make substaintial contribution to the academia. I suggest the authors to summarize a table or figure for each Chapter (Chapter 3 to 6) to show more direct comparison or key info in this part.
2. Oil sludge is a complex mixture with mostly solid, water and oil; and these phases can mix and form more complicated forms. Usually the treatment techniques are categorized according to a specific phase or a kind of material. I did not see this sortation in Chapter 5 and 6; now, it seems that everything is just listed together without an appropriate logic sequence. Moreover, I did not see the new achievement and frontier in oily sludge research clearly, for example, in line 462, why are rhamnolipids the biosurfactant that attracts more attention? Is it because it's biodegradable? Low cost and high efficiency? Which specific kind of waste is this biosurfactant suitable for? Are there quantitative data, say the oil elimination is improved from 90% to 95%? etc.
3. The journal is called Molecules. I was hoping that the authors could illustrate and discuss the context from a molecular point of view, so that the article is more appropriate to the journal.
4. Please mark 'taiwan' as 'taiwan (PRC)' or other appropriate terms in Fig. 3 to avoid uneccesary trouble to the authors and the journal.
Author Response
Please see the fil attached

Reviewer 3 Report
The authors of the work under consideration carried out a systematic review to examine and assess 379 journal articles on the development of oily sludge research that were obtained from the Web of Science database. The review paper provides some pertinent information in the field; therefore, the material may be of interest to the journal's readers. However, there are a few things to be checked before the manuscript is considered further:
· The entire second paragraph of the introduction has no citation; the presented information should somehow be cited.
· L46: It would be nice if the authors highlight some typical examples of petroleum hydrocarbons or heavy metals available in the oily sludge with their health and environmental implications.
· Oily sludge is a complex mixture of organic and inorganic metalloid chemicals that are resistant to natural degradation due to their structure and bioavailability. The authors should highlight that in the manuscript.
· It would have been more interesting if the authors had discussed the effectiveness of some explored wastewater treatment systems (within the studied period) in terms of the removal of some significant wastewater contaminants; especially in terms of removal efficiency.
· The authors should also check the possibility of improving the fonts in the figures; especially for figures 4, 5, and 6.
· L588: the term “web of science” should be capitalized.
Author Response
Please see the file attached

Round 2
Reviewer 2 Report
I appreciate the authors' careful effort in addressing my questions. I do not have any more comment on the manuscript.
Author Response
Reviewer’s Comment
I appreciate the authors' careful effort in addressing my questions. I do not have any more comment on the manuscript.
Authors’ Reply
Thank you
Reviewer 3 Report
The authors addressed most of the highlighted comments. The manuscript can now be considered for publication.
Author Response
Reviewer’s Comment
The authors addressed most of the highlighted comments. The manuscript can now be considered for publication.
Authors’ Reply
Thank you